# MgNO: Efficient Parameterization of Linear Operators via Multigrid

**Juncai He**,* **Xinliang Liu**\*†**& Jinchao Xu**
King Abdullah University of Science and Technology
{juncai.he, xinliang.liu, jinchao.xu}@kaust.edu.sa

## Abstract

In this work, we propose a concise neural operator architecture for operator learning. Drawing an analogy with a conventional fully connected neural network, we define the neural operator as follows: the output of the $i$-th neuron in a nonlinear operator layer is defined by $O_i(u) = \sigma\left(\sum_j \mathcal{W}_{ij} u + \mathcal{B}_{ij}\right)$. Here, $\mathcal{W}_{ij}$ denotes the bounded linear operator connecting $j$-th input neuron to $i$-th output neuron, and the bias $\mathcal{B}_{ij}$ takes the form of a function rather than a scalar. Given its new universal approximation property, the efficient parameterization of the bounded linear operators between two neurons (Banach spaces) plays a critical role. As a result, we introduce MgNO, utilizing multigrid structures to parameterize these linear operators between neurons. This approach offers both mathematical rigor and practical expressivity. Additionally, MgNO obviates the need for conventional lifting and projecting operators typically required in previous neural operators. Moreover, it seamlessly accommodates diverse boundary conditions. Our empirical observations reveal that MgNO exhibits superior ease of training compared to CNN-based models, while also displaying a reduced susceptibility to overfitting when contrasted with spectral-type neural operators. We demonstrate the efficiency and accuracy of our method with consistently state-of-the-art performance on different types of partial differential equations (PDEs).

## 1 Introduction

Partial differential equation (PDE) models are ubiquitous in physics, engineering, and other disciplines. Many scientific and engineering fields rely on solving PDEs, such as optimizing airfoil shapes for better airflow, predicting weather patterns by simulating the atmosphere, and testing the strength of structures in civil engineering. Tremendous efforts have been made to solve various PDEs arising from different areas. Solving PDEs usually requires designing numerical methods that are tailored to the specific problem and depend on the insights of the model. However, deep learning models have shown great promise in solving PDEs in a more general and efficient way. Recently, several novel methods, including Fourier neural operator (FNO) (Li et al., 2020), Galerkin transformer (GT) (Cao, 2021), deep operator network (DeepONet) (Lu et al., 2021), and convolutional neural operators Raonic et al. (2023), have been developed to directly learn the operator (mapping) between infinite-dimensional parameter and solution spaces of PDEs. These methods leverage advanced architectures, such as Fourier convolution and self-attentions, to handle a variety of input parameters and achieve high performance for forward and inverse PDE problems. The methodology has been applied to biomechanical engineering and weather forecasting (You et al., 2022; Pathak et al., 2022). Although deep neural operators have shown great potential to learn complex patterns and relationships from data, they still have some limitations compared to the classical numerical methods, such as lower accuracy and less flexibility to complex domain and boundary conditions. To address these challenges, we investigate the intrinsic properties of PDE-governed tasks and design the neural operator architecture based on multigrid methods, one of the most commonly used and efficient classical numerical algorithms.

---

*These authors contributed equally to this work.
†Corresponding author.

In this work, we propose a concise and elegant neural network architecture for operator learning. Our contributions are summarized as follows:

- We introduce a novel formulation for neural operators where the connections between neurons are characterized as bounded linear operators within function spaces. This approach yields a new universal approximation result, making the commonly used lifting and projecting operators in traditional constructs unnecessary.

- Central to this formulation is the efficient parameterization of linear operators. In response, we propose a distinct neural operator architecture, denoted MgNO. It employs the multigrid, with standard convolutions to parameterize the linear operator between neurons, aligning with our overarching goal of conciseness. Given the inherent ties between convolutions and multigrid methods in PDE contexts, MgNO naturally accommodates various boundary conditions.

- Our method establishes its superiority not only in prediction accuracy but also in efficiency, both in terms of parameter count and runtime on several PDEs, including Darcy, Helmholtz, and Navier-Stokes equations, with different boundary conditions. Such efficiency and precision resonate with the core philosophy of the multigrid method. Despite its minimal parameter count, our model exhibits rapid convergence without susceptibility to overfitting compared with other models, indicating a problem worth further investigation from both theoretical and practical perspectives.

## 2 Background and related work

**Neural operators** In recent times, substantial efforts have been dedicated to the development of neural networks or operators tailored for solving partial differential equations (PDEs). Lu et al. (2021) introduced the DeepONet, employing a branch-trunk architecture grounded in the universal approximation theorem in Chen et al. (1996). FNO parameterizes the global convolutional operators using a fast Fourier transform. Furthermore, geo-FNO Li et al. (2022) addresses tasks involving intricate geometries, such as point clouds, by transforming data into a latent uniform mesh and back. U-NO Rahman et al. (2022) improves FNO with a U-shaped architecture. The latent spectral model (LSM) (Wu et al., 2023) tried to identify the latent space and represent the operator in latent space. In addition, MWT Gupta et al. (2021) introduces a multiwavelet-based operator by parametrizing the integral operators using a fast wavelet transform. Then, Cao explored the self-attention mechanism Vaswani et al. (2017) and introduced a Galerkin-type attention mechanism with linear complexity for solving PDEs. More recently, a convolution-based architecture has been employed as a neural operator, as presented in Raonic et al. (2023). Among the notable approaches, spectral-type neural operators have gained attention for their global operation characteristics. The non-local operators have shown very promising results. Mathematical viewpoints provide universal approximation property for such operators (Kovachki et al., 2021; Lanthaler et al., 2023). However, their inherent limitation lies in their incapability to effectively handle boundary conditions. Methods like FNO, MWT, UNO, and LSM, rely on encoding boundary information within the input data, making them reliant on specific data representations. These methods tend to learn low-frequency component (Liu et al., 2022), which is not a surprise since high-frequency modes are often truncated for the spectral type method. In these methods, convolutions in the neural operator construction are parametrized by Fourier, or wavelet transforms. However, Fourier or wavelet-based methods are not always appropriate for solving PDEs because of boundary conditions and aliasing errors. In contrast, local and geometrical neural networks, such as ResNet, DilResNet, UNet He et al. (2016a;b); Ronneberger et al. (2015), offer greater flexibility in managing diverse boundary conditions. Nevertheless, these approaches were not originally designed for solving PDEs and often necessitated a substantial number of parameters to achieve high accuracy. Furthermore, they tend to act as high-pass filters, focusing primarily on local features like edges, surfaces, and textures, which limits their ability to capture long-distance relationships crucial in various physical phenomena. Large kernel convolution could mitigate the issue, but the design of large kernels requires hand-crafted since it is often hard to train (Ding et al., 2022).

**Multigrid** Multigrid methods Hackbusch (2013); Xu (1989); Trottenberg et al. (2000) stand out as some of the most efficient numerical approaches in scientific computing, especially when tackling elliptic PDEs. Intriguingly, within the deep learning community, multigrid methods first received

mention in the original ResNet paper He et al. (2016a), where the authors cited the methods as a core rationale behind the utility of residuals. Subsequently, the authors in He & Xu (2019); He et al. (2023) established deeper and more extensive links between multigrid methods and ResNet, giving rise to what is now known as MgNet. The research rigorously demonstrated that the linear V-cycle multigrid structure for the Poisson equation, can be represented as a convolutional neural network, despite their experimental focus on nonlinear multigrid with single-cycle structure for vision-related tasks. The original MgNet and its subsequent adaptations in the context of numerical PDEs Chen et al. (2022) and forecasting scenarios Zhu et al. (2023), have not provided a simple but effective solution in the realm of operator learning. In this work, we aim to fundamentally integrate multigrid methodologies with operator learning based on a very general framework.

## 3    A MORE GENERAL DEFINITION OF NEURAL OPERATORS

In this section, we present general deep artificial (abstract) fully connected (regarding neurons) neural operators as maps between two Banach function spaces $\mathcal{X} = H^s(\Omega)$ and $\mathcal{Y} = H^{s'}(\Omega)$ on a bounded domain $\Omega \subset \mathbb{R}^d$.

### 3.1    SHALLOW NEURAL OPERATORS FROM $\mathcal{X}$ TO $\mathcal{Y}$

To begin, we define the following shallow neural operators with $n$ neurons for operators from $\mathcal{X}$ to $\mathcal{Y}$ as

$$O(u) = \sum_{i=1}^{n} \mathcal{A}_i \sigma \left( \mathcal{W}_i u + \mathcal{B}_i \right) \quad \forall u \in \mathcal{X} \tag{1}$$

where $\mathcal{W}_i \in \mathcal{L}(\mathcal{X}, \mathcal{Y}), \mathcal{B}_i \in \mathcal{Y}$, and $\mathcal{A}_i \in \mathcal{L}(\mathcal{Y}, \mathcal{Y})$. Here, $\mathcal{L}(\mathcal{X}, \mathcal{Y})$ denotes the set of all bounded (continuous) linear operators between $\mathcal{X}$ and $\mathcal{Y}$, and $\sigma : \mapsto \mathbb{R}$ defines the nonlinear point-wise activation.

Despite the absence of commonly used lifting and projection layers , we still have the following universal approximation theorem based on this unified definition of shallow networks.

**Theorem 3.1** *Let $\mathcal{X} = H^s(\Omega)$ and $\mathcal{Y} = H^{s'}(\Omega)$ for some $s, s' \geq 1$, and $\sigma \in C(\mathbb{R})$ is non-polynomial, for any continuous operator $O^* : \mathcal{X} \mapsto \mathcal{Y}$, compact set $C \subset \mathcal{X}$ and $\epsilon > 0$, there is $n$ such that*

$$\inf_{O \in \Xi_n} \sup_{\boldsymbol{u} \in C} \|O^*(\boldsymbol{u}) - O(\boldsymbol{u})\|_{\mathcal{Y}} \leq \epsilon, \tag{2}$$

*where $\Xi_n$ denote the shallow networks defined in equation 1 with n neurons.*

**Sketch of the proof:**    We begin by approximating $O^*(u) \approx \sum_{i=1}^{m} f_i(u)\phi_i$, where this is viewed as a piecewise-constant operator. For this approximation, $\phi_i \in \mathcal{Y}$ and $f_i : \mathcal{X} \mapsto \mathbb{R}$ are continuous functionals for all $i = 1 : m$. Subsequently, we approximate $f_i(u)\phi_i$ by the aforementioned shallow neural operator $O_i(u)$ by using the properties of $\mathcal{X} = H^s(\Omega)$ and the classical universal approximation results for shallow neuron networks on $\mathbb{R}^k$. A detailed proof of this can be found in Appendix A.

### 3.2    DEEP NEURAL OPERATORS FROM $\mathcal{X}$ TO $\mathcal{Y}$

For clarity, let us define the product space $\mathcal{Y}^n := \underbrace{\mathcal{Y} \otimes \mathcal{Y} \otimes \cdots \otimes \mathcal{Y}}_{n}$. Subsequently, the bounded linear operator acting between these product spaces can be expressed as $\mathcal{W} \in \mathcal{L}(\mathcal{Y}^n, \mathcal{Y}^m)$. To be more specific, the relation $[\mathcal{W}\boldsymbol{h}]_i = \sum_{j=1}^{n} \mathcal{W}_{ij}\boldsymbol{h}_j$ holds for $i = 1 : n, j = 1 : m$, where $\boldsymbol{h}_j \in \mathcal{Y}$ and $\mathcal{W}_{ij} \in \mathcal{L}(\mathcal{Y}, \mathcal{Y})$. Now, let us denote $n_\ell \in \mathbb{N}^+$ for all $\ell = 1 : L$ as the number of neurons in $\ell$-th hidden layer with $n_{L+1} = 1$. Then, the deep neural operator with $L$ hidden layers and $n_\ell$ neurons in $\ell$-th layer is defined as

$$\begin{cases} \boldsymbol{h}^0 = \boldsymbol{u} \in \mathcal{X} \\ \boldsymbol{h}^\ell(\boldsymbol{u}) = \sigma \left( \mathcal{W}^\ell \boldsymbol{h}^{\ell-1}(\boldsymbol{u}) + \mathcal{B}^\ell \right) \in \mathcal{Y}^{n_\ell} \quad \ell = 1 : L \\ O(\boldsymbol{u}) = \mathcal{W}^{L+1} \boldsymbol{h}^L(\boldsymbol{u}) \in \mathcal{Y} \end{cases} \tag{3}$$

where $\mathcal{W}^\ell \in \mathcal{L}(\mathcal{Y}^{n_{\ell-1}}, \mathcal{Y}^{n_\ell})$ with $\mathcal{W}^1 \in \mathcal{L}(\mathcal{X}, \mathcal{Y}^{n_1})$ and $\mathcal{B}^\ell \in \mathcal{Y}^{n_\ell}$. Unless otherwise specified, for simplicity, in the rest of the article, we assume $n_\ell = n$.

We note that a standard hidden layer in GNO Kovachki et al. (2023) or FNO Li et al. (2020) is given by

$$h^\ell(u) = \sigma\left(\mathcal{W}^\ell h^{\ell-1}(u) + B^\ell h^{\ell-1}(u) + b^\ell\right) \in \mathcal{Y}^{n_\ell}, \tag{4}$$

where $\left[\mathcal{W}^\ell h^{\ell-1}(u)\right]_i(x) = \int_\Omega K_i(x, x') \cdot h^{\ell-1}(u)(x')dx'$, $B^\ell \in \mathbb{R}^{n \times n}$, and $b^\ell \in \mathbb{R}^n$. The term $B^\ell h^{\ell-1}$ in GNO or FNO is typically referred to as a local linear operator. In our approach, the term is interpreted as a specific parameterization of $\mathcal{B}^\ell$ in equation 3. To parameterize $\mathcal{B}^\ell \in \mathcal{Y}^n$, the task boils down to selecting appropriate bases for $\mathcal{Y}$. If we adopt the bases $\left\{h_1^{\ell-1}, \cdots, h_n^{\ell-1}, \mathbf{1}(x)\right\} \subset \mathcal{Y}$ for each $\left[\mathcal{B}^\ell\right]_i \in \mathcal{Y}$ with $i = 1, \ldots, n$, then we can express $\left[\mathcal{B}^\ell\right]_i = \sum_{j=1}^n B_{ij}^\ell h_j^{\ell-1} + b_i^\ell \mathbf{1}(x)$. Further, this can be compactly written in vector form as $\mathcal{B}^\ell = B^\ell h^{\ell-1} + b^\ell$, where $B^\ell \in \mathbb{R}^{n \times n}$ and $b^\ell \in \mathbb{R}^n$.

From equation 3, a notable feature of our framework is the elimination of the commonly used lifting layer in the first layer and the projection layer in the last layer, which are prevalent in most prior neural operators Kovachki et al. (2023); Li et al. (2022). It's important to highlight that these two artificial layers are considered crucial for demonstrating the approximation capabilities across a wide range of neural operators, as evidenced in Lanthaler et al. (2023).

However, in our proof of Theorem 3.1, a key insight that allows us to remove the lifting and projection layers is the high expressive capability of the linear operator $\mathcal{W}_{ij}^\ell \in \mathcal{L}(\mathcal{Y}, \mathcal{Y})$, which maps from the $i$-th input neuron to the $j$-th output neuron for all $i, j = 1 : n$. As a result, a primary contribution of this work is the introduction of a convolutional multigrid structure to parameterize $\mathcal{W}_{ij}^\ell$.

## 4 PARAMETRIZATION OF $\mathcal{W}_{ij}^\ell$ AND MGNO

In this section, we begin by discussing the rationale behind using multigrid to parameterize $\mathcal{W}_{ij}^\ell$. We then describe a standard multigrid process, framed in convolution language (single channel), for solving an elliptic PDE. Subsequently, we outline the global parameterization of $\mathcal{W}^\ell$ inspired by the multi-channel multigrid structure. Finally, we introduce our neural operator, MgNO, and touch upon its boundary-preserving characteristics.

**Motivation** Given the Schwartz kernel theorem Schwartz (1950 and 1951); Duistermaat et al. (2010), it is reasonable to select $\mathcal{W}_{ij}^\ell$ as a kernel version, i.e. $\mathcal{W}_{ij}^\ell u(x) = \int K_{i,j}(x, x')u(x')dx'$ as in equation 4. The primary challenge lies in appropriately parameterizing kernel functions $K_{i,j}(x, x')$ on $\Omega \times \Omega$. We further narrow our focus to a specific type of kernel known as Green's functions, which correspond to certain elliptic PDEs with specific boundary conditions. For practical purposes, we consider the discrete case for direct operator learning. Specifically, we have $\Omega = [0, 1]^d$ for $d = 1, 2, 3$ and

$$\mathcal{X} = \mathcal{Y} = \mathcal{V}_h(\Omega) := \text{linear finite element space on } \Omega \text{ with mesh size } h. \tag{5}$$

For instance, when $d = 2$, the input is discretized as $u(x) = \sum_{i,j=1}^d u_{ij}\phi_{i,j}(x) \mapsto u \in \mathbb{R}^{d \times d}$, where $d = \frac{1}{h}$. Consequently, for any $\mathcal{W}_{ij}^\ell \in \mathcal{L}(\mathcal{Y}, \mathcal{Y})$, the dimensionality of the space $\mathcal{L}(\mathcal{Y}, \mathcal{Y})$ becomes $d^2 \times d^2$. This dimensionality is challenging to parameterize directly for large values of $d$. Most existing neural operators, such as those in Kovachki et al. (2023); Lanthaler et al. (2023), primarily offer low-rank approximations of the kernel function in the spectral or wavelet domain. In this work, we advocate for using multigrid structures to directly parameterize within the spatial domain.

### 4.1 MULTIGRID METHODS FOR DISCRETE ELLIPTIC PDEs WITH BOUNDARY CONDITIONS

First, we offer a concise and practical overview of the rationale and methodology behind using multigrid techniques to parameterize Green's functions. Consider the elliptic PDEs given by $\mathcal{L}u(x) = f(x)$ defined over the domain $\Omega = (0, 1)^2$ and subject to Dirichlet, Neumann, or periodic boundary conditions. Employing a linear FEM discretization with a mesh size defined as $h = \frac{1}{d}$, the discretized system can be expressed as:

$$A * u = f, \tag{6}$$

where $u, f \in \mathbb{R}^{d \times d}$. Here, $*$ represents the standard convolutional operation for a single channel, complemented by specific padding schemes determined by the boundary conditions. The kernel

$A$, of dimensions $3 \times 3$, is dictated by the elliptic operator $\mathcal{L}$ in conjunction with the linear FEM. Consequently, the inverse operation of $A_*$ corresponds to the discrete Green's function associated with $\mathcal{L}$ under a linear FEM framework. As demonstrated in He & Xu (2019); He et al. (2021), the V-cycle multigrid approach for solving equation 6 can be precisely represented as a conventional convolutional neural network with one-channel.

We provide a concise overview of the essential components of multigrid structure in the language of convolution as an operator mapping from $f$ to $u$:

1. **Input ($f$) and Initialization**: Set $f^1 = f$ and initialize with $u^{1,0} = 0$.
2. **Iteration (Smoothing) Process**: The algorithm iteratively refines $u$ based on the relation:

$$u^{\ell,i} = u^{\ell,i-1} + B^{\ell,i} * \left( f^\ell - A^\ell * u^{\ell,i-1} \right), \tag{7}$$

   where $\ell = 1 : J$ and $i \leq \nu_\ell$.

3. **Hierarchical Structure via Restriction and Prolongation**: The superscript $\ell$ denotes the specific hierarchical level or grid within the multigrid structure. Specifically, using the residual, we restrict the input $f^\ell$ and the current state $u^\ell$ to a coarser level through convolution with a stride of 2:

$$f^{\ell+1} = R_\ell^{\ell+1} *_2 \left( f^\ell - A^\ell * u^\ell \right) \in \mathbb{R}^{d_{\ell+1} \times d_{\ell+1} \times n}, \quad u^{\ell+1,0} = 0. \tag{8}$$

   Subsequently, we apply the smoothing iteration as in equation 7 to derive the correction from the coarser level. The correction is then prolonged from the coarser to the finer level using a de-convolution operation $P_{\ell+1}^\ell$ with a stride of 2 (acting as the transpose of the restriction operation). After this prolongation, one can either opt to prolong further to an even coarser level (known as the Backslash-cycle) or proceed with post-smoothing followed by prolongation (referred to as the V-cycle).

Let's represent the linear operator defined by the aforementioned V-cycle multigrid operator as $\mathcal{W}_{Mg}$. The convergence result presented in Xu & Zikatanov (2002)

$$\| u - \mathcal{W}_{Mg}(A * u) \|_A \leq \left( 1 - \frac{1}{c} \right) \| u \|_A, \tag{9}$$

demonstrates the uniform approximation capabilities of $\mathcal{W}_{Mg}$ relative to the inverse of $A_*$, which corresponds to the Green's function associated with the elliptic operator $\mathcal{L}$. Here, $c$ is a constant that is independent of the mesh size $h$, and $\|u\|_A^2 = (u, A * u)_{L^2(\Omega)}$ denotes the energy norm. Please refer the Section C in Appendix for more details regarding the approximation property. We quantify the constant $c$ in equation 9 numerically using a concrete example.

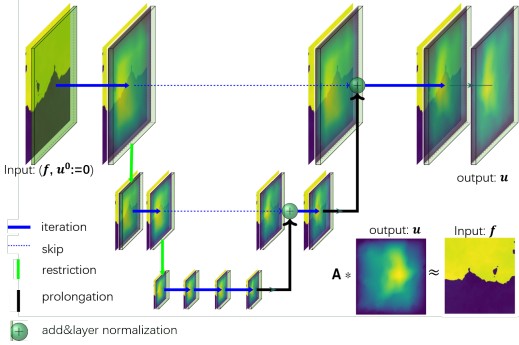

Figure 1: Overview of $\mathcal{W}_{Mg}$ using a multi-channel V-cycle multigrid framework.

## 4.2 Architecture of MgNO

Finally, we introduce a surrogate operator, denoted as MgNO. This operator maps from the linear finite element space of input functions, represented by $u \in \mathbb{R}^{d \times d \times c_{in}} \cong \mathcal{X} := \mathcal{V}_h(\Omega)$, to the linear

finite element space of output functions, represented by $\boldsymbol{v} \in \mathbb{R}^{d \times d \times c_{out}} \cong \mathcal{Y} := \mathcal{V}_h(\Omega)$. The mapping is defined as:

$$
\begin{cases}
\boldsymbol{h}^0 = \boldsymbol{u} \in \mathcal{X}, \\
\boldsymbol{h}^\ell(\boldsymbol{u}) = \boldsymbol{\sigma} \left( \mathcal{W}_{Mg}^\ell \boldsymbol{h}^{\ell-1}(\boldsymbol{u}) + \boldsymbol{B}^\ell \boldsymbol{h}^{\ell-1}(\boldsymbol{u}) + b^\ell \mathbf{1} \right) \in \mathcal{Y}^n, \quad \ell = 1 : L, \\
\boldsymbol{v} = \widetilde{\mathcal{G}}_\theta(\boldsymbol{u}) = \mathcal{W}_{Mg}^{L+1}(\boldsymbol{h}^L(\boldsymbol{u})) \in \mathcal{Y}.
\end{cases} \tag{10}
$$

Here, $\mathcal{W}_{Mg}^1 \in \mathcal{L}(\mathcal{X}, \mathcal{Y}^n)$, $\mathcal{W}_{Mg}^\ell \in \mathcal{L}(\mathcal{Y}^n, \mathcal{Y}^n)$, for $\ell = 2 : J$ and $W_{Mg}^{L+1} \in \mathcal{L}(\mathcal{Y}^n, \mathcal{Y})$ are multi-channel linear operators that process multi-channel input $\boldsymbol{h}^{\ell-1}$ and output $\mathcal{W}_{Mg}^\ell(\boldsymbol{h}^{\ell-1})$. The function $\sigma$ is the point-wise GELU activation, with $\boldsymbol{B}^\ell \in \mathbb{R}^{n \times n}$ and $b^\ell \in \mathbb{R}^n$.

We observe that directly using a one-channel $\mathcal{W}_{Mg}$ to parameterize each $\mathcal{W}_{ij}^\ell$ in equation 3 *individually* lead to expressivity and efficiency limitations. It is advantageous to design all $A^\ell$, $B^{\ell,i}$, restriction operators $R_\ell^{\ell+1}$, and prolongation operators $P_{\ell+1}^\ell$, for $\ell = 1 : J$, as multi-channel convolutional kernels. A detailed exposition of the multi-channel multigrid in the convolutional context is available in Algorithm 1 Appendix B and figure 1. Two primary motivations drive our choice of this strategy. Firstly, as established in He et al. (2022), channels in deep CNNs, for 2D inputs, function analogously to neurons in terms of their universal approximation capabilities. Consequently, augmenting the number of channels in multigrid enhances the expressiveness of $\mathcal{W}_{Mg}$. Secondly, this approach aligns with classical multigrid methods applied to solve elliptic partial differential equation system, including linear elasticity problems in 2D or 3D scenarios. In such cases, the V-cycle multigrid methods can be naturally represented as linear MgNO with multi-channel inputs. From a practical standpoint, the multi-channel variant of $\mathcal{W}_{Mg}$ possesses a parameter count on the order of $O\left(\log(d)n^2\right)$ and complexity on the order of $O(d^2n^2)$. Ultimately, by substituting $\mathcal{W}^\ell$ in equation 3, with $\mathcal{W}_{Mg}^\ell$, we derive our MgNO as presented in equation 10. It's important to note that $\mathcal{W}_{Mg}^\ell$ does not include any nonlinear activation.

**Boundary-preserving discretization in MgNO** It is imperative to highlight that MgNO is adept at accommodating the boundary conditions of various PDEs. Leveraging the inherent relationship between convolutions and multigrid, the output of $\mathcal{W}_{Mg}^\ell$ can seamlessly satisfy Dirichlet, Neumann, or periodic boundary conditions. Consequently, $\widetilde{\mathcal{G}}_\theta(u)$ can maintain these boundary conditions without any training phase, provided by taking $\boldsymbol{B}^1 = 0$ in equation 10 and ensuring $\mathbf{1}(x)$ satisfies the same boundary conditions. One can refer to Table 4 in Section F for specific convolution configuration for specific boundary conditions, and also Section C in Appendix for an concrete example of integrating boundary conditions into neural operators.

## 5 EXPERIMENTS

**General setting** We consider pairs of functions $(\boldsymbol{u}_j, \boldsymbol{v}_j)_{j=1}^N$, where $\boldsymbol{u}_j$ is drawn from a probability measure $\mu$ and $\boldsymbol{v}_j = \mathcal{G}(\boldsymbol{u}_j)$. Given the data $(\boldsymbol{u}_j, \boldsymbol{v}_j)_{j=1}^N$, we approximate $\mathcal{G}$, by solving the network parameter set $\theta$ via an optimization problem:

$$
\min_{\theta \in \Theta} \mathcal{L}(\theta) := \min_{\theta \in \Theta} \frac{1}{N} \sum_{j=1}^N \left[ \|\widetilde{\mathcal{G}}_\theta(\boldsymbol{u}_j) - \boldsymbol{v}_j\|^2 \right]. \tag{11}
$$

**Benchmarks** To substantiate the superiority of our method, we conducted comprehensive method comparisons across multiple benchmark scenarios. These benchmarks were categorized based on their association with specific partial differential equations (PDEs, namely Darcy, Navier-Stokes, and Helmholtz benchmarks. Furthermore, we incorporated tasks encompassing both regular and irregular domains, each subject to a variety of boundary conditions. This comprehensive comparison allows us to thoroughly assess the robustness and adaptability of our method, ensuring its effectiveness in real-world scenarios. For clearness, we summarize the benchmarks in Table 5. Please also refer to Section D for more detailed descriptions.

**Baselines** We perform a comprehensive comparison with existing methods: (1) We include the original FNO (Li et al., 2020); (2) UNet (Ronneberger et al., 2015) and U-NO (Rahman et al.,

2022); (3) multiwavelet neural operator (MWT) (Gupta et al., 2021); (4) Galerkin transformer (GT) (Cao, 2021); (5) latent spectral models (LSM) (Wu et al., 2023); (6) learned simulator for turbulence (DilResNet) (Stachenfeld et al., 2021). (7) Fine-tuned FNO (sFNO-v2) (Benitez et al., 2023). The baseline models were implemented using their official implementations. For consistency, we executed the baselines with default hyperparameters unless the experiments' details specified otherwise. In cases where specific experiment details were lacking in the literature, we employed a reasonable network scale to ensure a fair comparison. In our training setup, we implemented a comprehensive approach. We trained the baselines using multiple configurations, including their default training settings encompassing different loss functions, training algorithms, and schedulers. We then reported the best results from these trials to mitigate any training-related variations. Notably, our observations revealed that all models benefited from the scheduler utilizing cosine annealing learning rates. Figure 3 will visually demonstrate that all models underwent sufficient training, ensuring that their performance accurately reflects their capabilities.

**Darcy**   The Darcy problems, widely recognized for their applicability to multigrid methods, serve as a key focus. We anticipate substantial performance improvements compared to the current state-of-the-art. To thoroughly assess our method, we incorporated three distinct Darcy benchmarks: Darcy smooth, Darcy rough in Li et al. (2020), and Darcy multiscale in Liu et al. (2022). Darcy smooth and Darcy rough both present two-phase media within the domain, featuring different interface roughness levels. In contrast, Darcy's multiscale presents media with multiple scales of permeability. Darcy rough and multiscale Darcy challenges our method's ability to capture operator dependencies on fine-scale features, a task that poses difficulties for existing methods. Please refer D.1 for detailed datasets description and experiments setup.

Table 1: Performance comparison for Darcy benchmarks. Performance are measured with relative $L^2$ errors ($\times 10^{-2}$) and relative $H^1$ errors ($\times 10^{-2}$).

| Model | time (s/iter) | params (m) | Darcy smooth | | Darcy rough | | Darcy multiscale | |
|---|---|---|---|---|---|---|---|---|
| | | | $L^2$ | $H^1$ | $L^2$ | $H^1$ | $L^2$ | $H^1$ |
| FNO2D | 7.4 | 2.37 | 0.684 | 2.583 | 1.613 | 7.516 | 1.800 | 9.619 |
| DilResNet | 14.9 | 1.04 | 4.104 | 5.815 | 7.347 | 12.44 | 1.417 | 3.528 |
| UNet | 9.1 | 17.27 | 2.169 | 4.885 | 3.519 | 5.795 | 1.425 | 5.012 |
| U-NO | 11.4 | 16.39 | 0.492 | 1.276 | 1.023 | 3.784 | 1.187 | 5.380 |
| MWT | 21.7 | 9.80 | — | — | 1.138 | 4.107 | 1.021 | 7.245 |
| GT | 38.2 | 2.22 | 0.945 | 3.365 | 1.790 | 6.269 | 1.052 | 8.207 |
| LSM | 18.2 | 4.81 | 0.601 | 2.610 | 2.658 | 4.446 | 1.050 | 4.226 |
| MgNO | **6.6** | **0.57** | **0.176** | **0.576** | **0.339** | **1.380** | **0.715** | **1.756** |
| MgNO-high-in | 9.5 | 0.85 | 0.103 | 0.469 | 0.275 | 0.799 | 0.504 | 1.579 |

— MWT (Gupta et al., 2021) only supports resolution with powers of two.
— FNO2D, U-NO and MWT's performance are further improved from originally reported because of the usage of $H^1$ loss and scheduler.
— The runtime and number of parameters count are using Darcy rough as the example case.

**Remark 5.1** *Note that* MgNO-high-in *is not directly comparable with other models. The* MgNO-high-in *model differs from others, utilizing a higher input resolution of* $512 \times 512$ *in the Darcy rough case, compared to the standard* $256 \times 256$ *used in other models. Inspired by reduced-order modeling (Lucia et al., 2004; Hou et al., 1999; Engquist & Souganidis, 2008; Målqvist & Peterseim, 2014; Owhadi & Zhang, 2007), it represents high-resolution input but targets a solution in a lower dimensional space. This is achieved by adjusting the depth/level of the* $\mathcal{W}_{Mg}$ *that condenses the input to the desired output resolution. Our findings suggest that high-resolution inputs enhance fine-scale predictions, implying the model effectively balances computational efficiency with high fidelity. On the other hand, by incorporating appropriate finite element basis functions, MgNO, compatible with the finite element method, demonstrates the discretization invariant capability. MgNO can train on lower-resolution datasets and evaluate on higher resolutions without the necessity of high-resolution training data, thus accomplishing zero-shot super-resolution. Please see E for the detailed experiments.*

Table 5 and Figure 2 illustrate the precision with which our method predicts the fine-scale features of the solution. The qualitative comparisons of the contour lines in the top row indicate the model's aptitude in accurately capturing small-scale variations, evidenced further by the low $H^1$ error reported in Table 5. Moreover, our method establishes its superiority not only in prediction accuracy

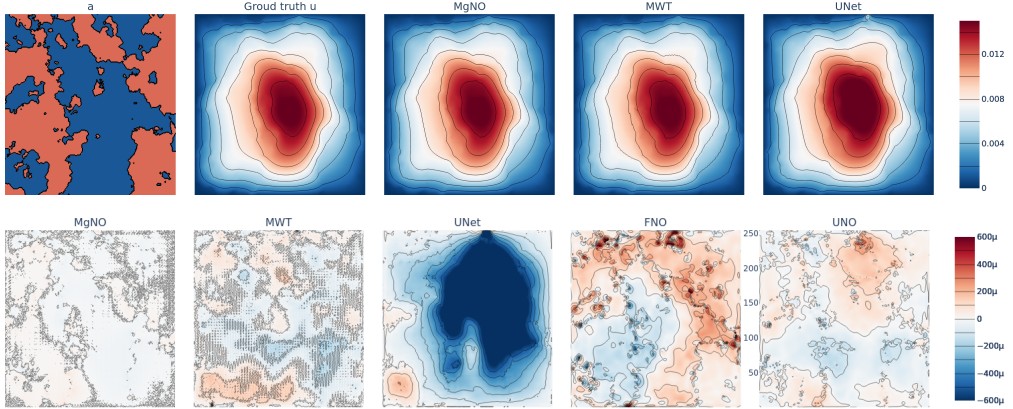

Figure 2: Qualitative comparisons on Darcy rough benchmark. Top: coefficient $a$, ground truth $u$, and predictions; bottom: the corresponding prediction error map for each model in the same color scale.

but also in efficiency, both in terms of parameter count and runtime. Such efficiency and precision resonate with the core philosophy of the multigrid method.

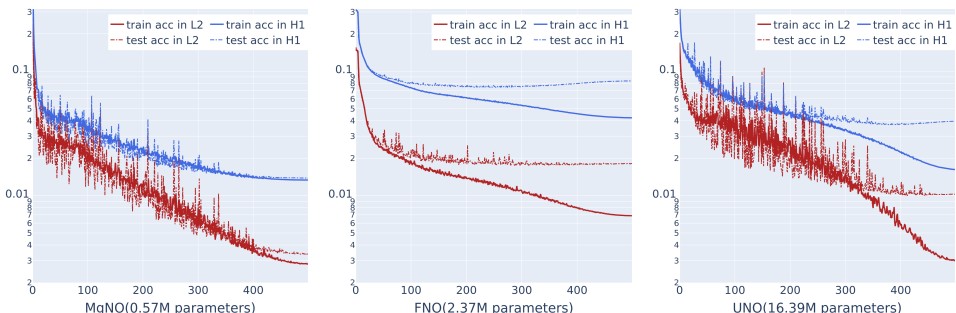

Figure 3: Comparison of training dynamics between MgNO, FNO and UNO. The x-axis represents the number of epochs, and the y-axis is the error in the log scale. We present both the $L^2$ and $H^1$ training and testing accuracy (errors). For full comparisons, please refer D.1

**Training Dynamics** In Figure 3, we present the training dynamics of different models on benchmark Darcy rough. We train all the models in the same setting. Please see D.1 for further training settings). Despite its minimal parameter count, our model exhibits rapid convergence without susceptibility to overfitting. It is noteworthy that while larger models like MWT(9.80M), UNO(16.39M), and LSM(4.81M) can achieve training errors comparable to, or even lower than, our model, their testing errors are considerably higher. This discrepancy underscores our model's robust generalization capabilities on unseen data.

**Helmholtz** We conduct experiments on the Helmholtz equation. The Helmholtz equation poses significant challenges for numerical methods, primarily due to the resonance phenomenon. Please refer to D.3 for more details about the dataset. We present the comparison in Table 3 and Figure 8.

**Navier Stokes** Numerous studies have explored the use of neural operators for fluid dynamics simulation Kochkov et al. (2021); Li et al. (2020); Stachenfeld et al. (2021); Mi et al. (2023). Our focus is on the 2D Navier-Stokes equation in its vorticity form defined over the unit torus, $\mathsf{T}$ (refer to Section D.2 for an in-depth discussion). The vorticity, represented by $\omega(x, t)$, is defined for $x \in \mathsf{T}$ and $t \in [0, T]$. We aim to learn the operator $\mathcal{S} : w(\cdot, 0 \le t \le 9) \to w(\cdot, 10 \le t \le T)$, which projects the vorticity up to time 9 onto the vorticity up to a later time $T$, following the benchmark set by Li et al. (2020). Our experiments consider viscosities of $\nu = 1e - 5$ and conclude at time $T = 20$.

Furthermore, we also test models on the same Navier-Stokes task with $\nu = 1e - 5$ but introduce an alternative training configuration labeled as "NS2". It is imperative to note that in the Navier Stokes $\nu = 1e - 5$ configuration with training tricks, all models incorporate these specialized training techniques. These modifications consistently enhance performance when compared to the original setup. The inclusion of these training tricks contributes to more stable generalization errors across various models, justifying their use in performance comparisons. Hence, both evaluation methods offer an equitable basis for contrasting the efficacy of our approach with existing baselines. A comprehensive overview of the training configurations and specific tricks can be found in Section D.2.

Table 2: Performance on Navier Stokes

| Model | time | params(m) | Pipe | NS | NS2 |
|---|---|---|---|---|---|
| FNO2D | **2.0** | **0.94** | 0.67 | 15.56 | 4.57 |
| UNET | 7.6 | 17.27 | 0.65 | 19.82 | 10.08 |
| DILRESNET | 5.9 | 1.04 | 0.31 | 23.28 | 10.91 |
| U-NO | 56.1 | 30.48 | 1.00 | 17.13 | 2.64 |
| MWT | 6.2 | 9.80 | 0.77 | 15.41 | 3.09 |
| LSM | 24.3 | 4.81 | 0.50 | 15.35 | 12.93 |
| MGNO | 6.4 | 2.89 | **0.24** | **10.82** | **1.63** |

Table 3: Performance on Helmholtz

| Model | time | params(m) | $L^2(\times 10^{-2})$ | $H^1(\times 10^{-2})$ |
|---|---|---|---|---|
| FNO2D | 5.1 | 1.33 | 1.69 | 9.92 |
| UNET | 7.1 | 17.26 | 3.81 | 23.31 |
| DILRESNET | 10.8 | 1.03 | 4.34 | 34.21 |
| U-NO | 21.5 | 16.39 | 1.26 | 8.03 |
| sFNO-v2 | 30.1 | 12.0 | 1.72 | 10.40 |
| LSM | 28.2 | 4.81 | 2.55 | 10.61 |
| MGNO | 15.1 | 7.58 | **0.71** | **4.02** |

Table 4: Ablation & hyperparameter study

| Model Configuration | $L^2$ Error ($\times 10^{-2}$) |
|---|---|
| MGNO, 4 levels | 2.10 |
| MGNO, 3 levels | 2.44 |
| MGNO, without boundary condition | 3.18 |
| MGNO, 6 layers | 1.47 |
| MGNO-FNO | 3.91 |
| **Baseline** MGNO | 1.63 |

**Ablation & Hyperparameter Study** The ablation study uses NS2 as a reference example. The baseline MgNO is configured with 5 levels, 32 channel dimensions, utilizes periodic boundary conditions, and comprises 5 architectural layers. One can find the detailed configuration in Table 7. We conduct the ablation& hyperparameters study by changing one model configuration and fixing the others. In the first and second rows, our analysis evaluated the impact of the number of levels (depth) within the MgNO architecture. Given that our structure simply merges convolution with a multigrid touch, this hyperparameter examination doubles as an ablation study for the influence of neural network depth. The outcomes highlight the performance sensitivity of MgNO to its depth. This mirrors properties observed in traditional multigrid methods where deeper algorithms are requisite for effectively mitigating low-frequency component errors. The third row emphasizes the role of boundary conditions in convolution operations, underscoring the distinct advantages of our method. Fourth row indicate adding more layers improves performance, but it also increases computational costs. The MGNO-FNO configuration offers a higher-level ablation. Here, we substituted the multigrid parameterized $\mathcal{W}_{i,j}$ with the spectral convolution detailed in Li et al. (2020). The ablation study reaffirms the strength of the MGNO approach, especially when integrating boundary conditions. This method not only resonates with traditional multigrid properties but also encapsulates the essence of convolution-based neural operators, delivering optimal performance in comparison to alternative convolution-based configurations.

## CONCLUSION

In this work, we introduced a novel formulation for neural operators, where neuron connections are characterized as bounded linear operators within function spaces, eliminating the need for traditional lifting and projecting operators. Central to our approach is the MgNO architecture, which leverages multigrid structure and its multi-channel convolutional form to efficiently parameterize linear operators, naturally accommodating various boundary conditions. Empirically, MgNO has demonstrated superior performance in both accuracy and efficiency across several PDEs, including Darcy, Helmholtz, and Navier-Stokes equations.

Moreover, there are still some questions worth future exploration. Our work does not sufficiently benchmark tasks on irregular domains. For irregular domains, our current implementation of MgNO requires a deformation mapping to transform the data into a regular format as shown in the pipe task. Therefore, incorporating algebraic multigrid methods Xu & Zikatanov (2017) will enhance the MgNO's applicability to inputs and outputs in general format such as arbitrary sampling data points and point clouds. Furthermore, MgNO achieves rapid convergence without overfitting, highlighting a promising direction for future research.

**Reproducibility Statement**  The complete code used in our experiments is available at `https://github.com/xlliu2017/MgNO/`. This repository includes all scripts, functions, and necessary files for reproducing our results. Datasets employed in our experiments can also be accessed via URLs provided in the repository. We have disclosed all configurations, encompassing hyperparameters, model architectures, and optimization strategies. This ensures that our experimental setup is fully transparent and can be replicated by others. To ensure the robustness of our results, each experiment was conducted at least three times with different runs initiated using unique random seeds. This approach is aimed at accounting for variability and guaranteeing the reliability of our findings.

**Acknowledgement**  This work was partially supported by the KAUST Baseline Research Fund.

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

# A  Proof of Theorem 3.1

Here we present the detailed proof of Theorem 3.1 with the following steps.

1. Piecewise constant approximation of $O^*$: Since $C \subset X$ is a compact set, for any $\epsilon > 0$, there are $\{\phi_1, \cdots, \phi_m\} \subset \mathcal{Y}$ and continuous functionals $f_i : X \mapsto \mathbb{R}$ for $i = 1 : m$ such that

$$\sup_{u \in C} \left\| O^*(u) - \sum_{i=1}^{m} f_i(u)\phi_i \right\|_{\mathcal{Y}} \le \frac{\epsilon}{3}. \tag{12}$$

Thus, we only need to prove that there is $O_i \in \Xi_{n_i}$ such that

$$\sup_{u \in C} \|f_i(u)\phi_i - O_i(u)\|_{\mathcal{Y}} \le \frac{\epsilon}{3m}. \tag{13}$$

2. Parameterization (approximation) of $X$ with finite dimensions to discretize $f_i$: Since $X = H^s(\Omega)$ and $C$ is compact, we can find $k \in \mathbb{N}^+$ such that

$$\sup_{u \in C} \left| f_i(u)\phi_i - f_i\left( \sum_{j=1}^{k} \left(u, \varphi_j\right)\varphi_j \right)\phi_i \right|_{\mathcal{Y}} \le \frac{\epsilon}{3m} \quad \forall i = 1 : m, \tag{14}$$

where $\varphi_i$ are the orthogonal basis in $H^s(\Omega)$. Then, for a specific $f_i : X \mapsto \mathbb{R}$, let us define the following finite-dimensional continuous function $F_i : \mathbb{R}^k \mapsto \mathbb{R}$ as

$$F_i(x) = f_i\left( \sum_{i=1}^{k} x_i\varphi_i \right), \quad \forall c \in [-M, M]^k, \tag{15}$$

where $M := \sup_i \sup_{u \in C}(u, \varphi_i)$.

3. Universal approximation of $F_i$ on $[-M, M]^k$ using classical shallow neural networks $\widetilde{F}_i(x)$ for $x \in [-M, M]^k$: If $\sigma : \mathbb{R} \mapsto \mathbb{R}$ is not polynomial, given by the results in Leshno et al. (1993), there is $n_i \in \mathbb{N}^+$ and

$$\widetilde{F}_i(x) = \sum_{j=1}^{n_i} a_{ij}\sigma(w_{ij} \cdot x + b_{ij}), \tag{16}$$

where $w_{ij} \in \mathbb{R}^k$ and $a_{ij}, b_{ij} \in \mathbb{R}$, such that

$$\sup_{x \in [-M, M]^k} \left| F_i(x) - \widetilde{F}_i(x) \right| \le \frac{\epsilon}{3m\|\phi_i\|_{\mathcal{Y}}}. \tag{17}$$

4. Representation of $\widetilde{F}_i(x)$ using $\Xi_{n_i}$: Now, let us define $O_i \in \Xi_{n_i}$ as

$$O_i(u) = \sum_{j=1}^{n_i} \mathcal{A}_{ij}\sigma\left( \mathcal{W}_{ij}u + \mathcal{B}_{ij} \right) \tag{18}$$

where

$$\mathcal{W}_{ij}u = w_{ij} \cdot \left((u, \varphi_1), \cdots, (u, \varphi_k)\right) \mathbf{1}(x) \in \mathcal{Y} \tag{19}$$

and $\mathcal{B}_{ij} = b_{ij}\mathbf{1}(x) \in \mathcal{Y}$, and

$$\mathcal{A}_{ij}(v) = a_{ij}\frac{(v, \mathbf{1})}{\|\mathbf{1}\|^2}\phi_i. \tag{20}$$

Here, $\mathbf{1}(x)$ denotes the constant function on $\Omega$ whose function value is 1. This leads to

$$\begin{aligned} O_i(u) &= \widetilde{F}_i\left(((u, \varphi_1), \cdots, (u, \varphi_k))\right)\phi_i \\ &\approx F_i\left(((u, \varphi_1), \cdots, (u, \varphi_k))\right)\phi_i \\ &= f_i\left( \sum_{j=1}^{k} \left(u, \varphi_j\right)\varphi_j \right)\phi_i. \end{aligned} \tag{21}$$

5. Triangle inequalities to finalize the proof:

$$\sup_{u \in C} \left\| O^*(u) - \sum_{i=1}^{m} O_i(u) \right\|_{\mathcal{Y}} \le \epsilon, \tag{22}$$

where $\sum_{i=1}^{m} O_i(u) \in \Xi_N$ with $N = \sum_{i=1}^{m} n_i$.

## B    Multigrid in convolution language

---

**Algorithm 1** $u = \mathcal{W}_{Mg}(f, u^{1,0}; J, \nu_\ell, n)$

---

1: **Input**: discretized input function $f \in \mathbb{R}^{d \times d \times c_f}$, discretized initial output function $u^{1,0} \in \mathbb{R}^{d_1 \times d_1 \times n}$, number of grids $J$, number of smoothing iterations $\nu_\ell$ for $\ell = 1 : J$, number of channels $n$ on each grid.

2: **Initialization**: $f^1 = K^0 * f \in \mathbb{R}^{d_1 \times d_1 \times n}$, $u^{1,0} \in \mathbb{R}^{d_1 \times d_1 \times n}$, $u^{1,0} = 0$ if $u^{1,0}$ is not given as input, and discretized spatial size $d_\ell = \frac{d}{2^{\ell-1}}$ for $\ell = 1 : J$.

3: **for** $\ell = 1 : J$ **do**

4:     Feature extraction (smoothing):

5:     **for** $i = 1 : \nu_\ell$ **do**

6:
$$u^{\ell,i} = u^{\ell,i-1} + B^{\ell,i} * \left( f^\ell - A^\ell * u^{\ell,i-1} \right) \in \mathbb{R}^{d_\ell \times d_\ell \times n}. \tag{23}$$

7:     **end for**

8:     Note: $u^\ell = u^{\ell,\nu_\ell}$

9:     **if** $\ell < J$ **then**

10:         Interpolation and restriction:
$$u^{\ell+1,0} = 0 \quad \text{and} \quad f^{\ell+1} = R_\ell^{\ell+1} *_2 \left( f^\ell - A^\ell * u^\ell \right) \in \mathbb{R}^{d_{\ell+1} \times d_{\ell+1} \times n}$$

11:     **end if**

12: **end for**

13: **for** $\ell = J - 1 : 1$ **do**

14:     Prolongation:
$$u^{\ell,0} = u^\ell + P_{\ell+1}^\ell *^2 u^{\ell+1} \in \mathbb{R}^{d_\ell \times d_\ell \times c}$$

15:     **for** $i = 1 : \nu_\ell$ **do**

16:         Skip if Backslash-cycle OR do the following post-smoothing if V-cycle:
$$u^{\ell,i} = u^{\ell,i-1} + B^{\ell,i} * \left( f^\ell - A^\ell * u^{\ell,i-1} \right) \in \mathbb{R}^{d_\ell \times d_\ell \times n}.$$

17:     **end for**

18: **end for**

19: **Output:**
$$u = u^{1,\nu_1} \in \mathbb{R}^{d \times d \times n}.$$

---

## C    A Numerical Example Illustrating the Approximation Property of $\mathcal{W}_{Mg}$ with Sepcific Boundary Condition in equation 9

We consider elliptic PDEs described by $-\Delta u(x) = f(x)$ over the domain $\Omega = (0,1)^2$, subject to Dirichlet boundary conditions where $u = 0$ on $\partial\Omega$. Using a linear Finite Element Method (FEM) discretization with mesh size $h = \frac{1}{(d+1)}$, the discretized equation is represented as:

$$A * u = f, \tag{24}$$

where $u, f \in \mathbb{R}^{d \times d}$ and $A = \begin{pmatrix} 0 & -1 & 0 \\ -1 & 4 & -1 \\ 0 & -1 & 0 \end{pmatrix}$. Here, $*$ denotes the standard convolution operation for a single channel, incorporating a zero-padding scheme with padding size 1 corresponding to the Dirichlet boundary condition.

In the one-channel $\mathcal{W}_{Mg}$ configuration, as detailed in Algorithm 1, we need further configurations to incorporating the Dirichlet boundary conditions. The convolutional layers $A^\ell$ and $B^{\ell,i}$ utilize zero-padding of size 1. In contrast, the layers $R\ell^{\ell+1}$ and $P_{\ell+1}^\ell$, for $\ell = 1 : J$, are set up without any

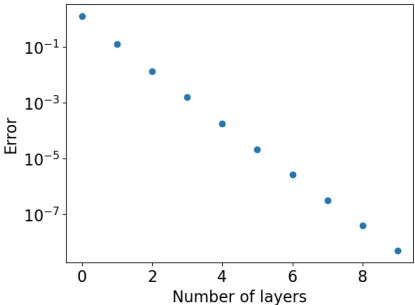

Figure 4: The error, quantified on a logarithmic scale, numerically demonstrates the approximation rate, which is approximately $1 - \frac{1}{c} \approx 0.1$, as outlined in equation 9.

padding. The convolution weights are configured as follows:

$$A^\ell = \begin{pmatrix} 0 & -1 & 0 \\ -1 & 4 & -1 \\ 0 & -1 & 0 \end{pmatrix}, \ B^{\ell,i} = \begin{pmatrix} 0 & 1/64 & 0 \\ 1/64 & 12/64 & 1/64 \\ 0 & 1/64 & 0 \end{pmatrix},$$

$$R^{\ell+1}_\ell = \begin{pmatrix} 0 & 1/2 & 1/2 \\ 1/2 & 1 & 1/2 \\ 1/2 & 1/2 & 0 \end{pmatrix}, \text{ and } P^\ell_{\ell+1} = \begin{pmatrix} 0 & 1/2 & 1/2 \\ 1/2 & 1 & 1/2 \\ 1/2 & 1/2 & 0 \end{pmatrix}.$$

Implementing $\mathcal{W}_{Mg}$ with these specific weights, we apply it iteratively to the right-hand side $f$ for $l$ iterations, analogous to $l$ layers in the $\mathcal{W}_{Mg}$ network. We define $u^{(l+1)} := \mathcal{W}_{Mg}(f, u^{(l)})$ with $u^{(0)} := 0$. The convergence rate observed is illustrated in Figure 4.

The configurations of these weights are based on fundamental facts of piecewise linear finite elements and multigrid algorithms. For further details on these configurations, one can refer to Briggs et al. (2000); Braess (2007). For Neumann boundary conditions and periodic boundary conditions, the implementation of reflect padding and periodic padding, respectively, each with a padding size of 1, aligns seamlessly with the discretization of the finite element method, facilitating straightforward application without essential difficulty. The convergence behaviors are the same.

## D  FURTHER DETAILS ON BENCHMARKS

| Benchmarks | Time dependent | Regular domain | # Dimension | BD | High resolution | Multi-channel input |
|---|---|---|---|---|---|---|
| DARCY SMOOTH | No | Yes | 2D | Dirichlet | No | No |
| DARCY ROUGH | No | Yes | 2D | Dirichlet | Yes | No |
| DARCY MULTISCALE | No | Yes | 2D | Dirichlet | Yes | No |
| NAVIER STOKES | Yes | Yes | 2D | Periodic | No | Yes |
| PIPE | No | No | 2D | Mixed | No | Yes |
| HELMHOLTZ | No | Yes | 2D | Neumann | No | No |

Table 5: Summary of benchmarks

### D.1  DARCY

The Darcy equation writes

$$\begin{cases} -\nabla \cdot (a(x)\nabla u(x)) = f(x) & x \in D \\ u(x) = 0 & x \in \partial D \end{cases} \tag{25}$$

where the coefficient $0 < a_{\min} \le a(x) \le a_{\max}, \forall x \in D$, and the forcing term $f \in H^{-1}(D; \mathbb{R})$. The coefficient to solution map is $\mathcal{S} : L^\infty(D; \mathbb{R}_+) \to H^1_0(D; \mathbb{R})$, such that $u = \mathcal{S}(a)$ is the target operator.

**Two-Phase Coefficient (Darcy smooth and Darcy rough)**    The two-phase coefficients and solutions (referred to as Darcy smooth and Darcy rough in Table 5) are generated according to `https://`

`github.com/zongyi-li/fourier_neural_operator/tree/master/data_generation`, and used as an operator learning benchmark in (Li et al., 2020; Gupta et al., 2021; Cao, 2021). The coefficients $a(x)$ are generated according to $a \sim \mu := \psi_\# \mathcal{N}\left(0, (-\Delta + cI)^{-2}\right)$ with zero Neumann boundary conditions on the Laplacian. The mapping $\psi : \mathbb{R} \to \mathbb{R}$ takes the value $a_{\max}$ on the positive part of the real line and $a_{\min}$ on the negative part. The push-forward is defined in a pointwise manner. The forcing term is fixed as $f(x) \equiv 1$. Solutions $u$ are obtained by using a second-order finite difference scheme on a suitable grid. The parameters $a_{\max}$ and $a_{\min}$ can control the contrast of the coefficient. The parameter $c$ controls the roughness (oscillation) of the coefficient; a larger $c$ results in a coefficient with rougher two-phase interfaces, as shown in Figure 2.

**Darcy multiscale coefficient**  We examine equation 25 that features a multiscale trigonometric coefficient adapted from (Owhadi, 2017). This corresponds to the Darcy multiscale benchmark highlighted in Table 5. The domain, denoted as $D$, spans the area $[-1, 1]^2$. The coefficient $a(x)$ is formulated as

$$a(x) = \prod_{k=1}^{6} \left(1 + \frac{1}{2}\cos(a_k \pi (x_1 + x_2))\right)\left(1 + \frac{1}{2}\sin(a_k \pi (x_2 - 3x_1))\right),$$

where each $a_k$ is uniformly distributed between $2^{k-1}$ and $1.5 \times 2^{k-1}$. The forcing term is consistently set to $f(x) \equiv 1$. Reference solutions are derived using $\mathcal{P}_1$ FEM on a grid resolution of $1023 \times 1023$. Refer to Figure 5 for visual representations of both the coefficient and the solution.

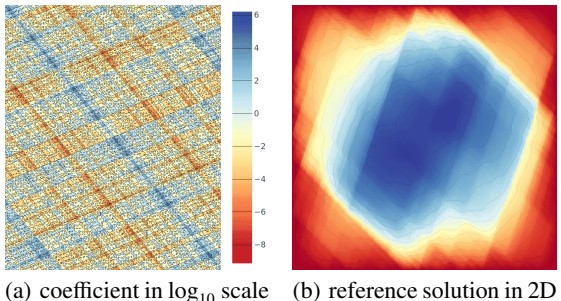

(a) coefficient in $\log_{10}$ scale    (b) reference solution in 2D

Figure 5: (a) multiscale trigonometric coefficient, (b) reference solution.

**Training settings**  In the Darcy rough scenario, our data comprises 1280 training, 112 validation, and 112 testing samples. For the Darcy smooth and multiscale trigonometric cases, the split is 1000 training, 100 validation, and 100 testing samples. Training is limited to 500 epochs for Darcy smooth and rough, and 300 epochs for Darcy multiscale.

We trained various baselines using multiple configurations, focusing primarily on default training settings with varying loss functions and schedulers. The optimal results were selected to ensure consistency. The consistent boost in performance across all models was observed using the $H^1$ loss and the OneCycleLR scheduler with cosine annealing. Specifically, optimal models are trained with a batch size of 8, the Adam optimizer, and OneCycleLR with cosine annealing. Although learning rates varied to optimize training across models, MgNO started with a rate of $5 \times 10^{-4}$, decreasing to $2.5 \times 10^{-6}$. FNO, UNO, and LSM default to $1 \times 10^{-3}$, with a weight decay of $1 \times 10^{-4}$. CNN-based models, like UNet and DilResNet, have a maximum rate of $5 \times 10^{-4}$. As indicated in Figure 3, a thorough training process ensured equitable model comparison.

All experiments were executed on an NVIDIA A100 GPU.

**Training dynamics**  We present the training dynamics of different models on benchmark Darcy rough. We train all the models in the same setting (we use the same batch size 8, Adam optimizer and OneCycleLR scheduler with cosine annealing. OneCycleLR scheduler with a small learning rate at the end allows models to be trained sufficiently. Note that the learning rate varies for different models to allow models to be trained sufficiently) for 500 epochs and record the history of training and testing errors during the process.

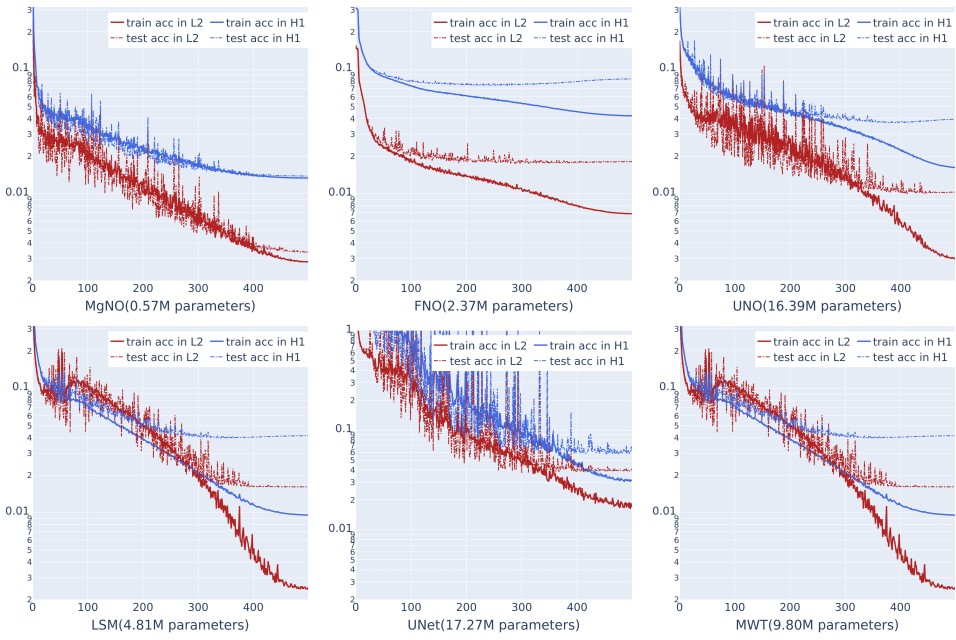

Figure 6: Comparison of training dynamics between MgNO, FNO and UNO. The x-axis represents the number of epoch and the y-axis the error in log scale. We present both the $L2$ and $H1$ training and testing accuracy (errors).

### D.2 NAVIER STOKES

We take the dataset to simulate incompressible and viscous flow on the unit torus, where the density of the fluid is unchangeable ($\rho$ in Eq. In this situation, energy conservation is independent of mass and momentum conservation. Hence, the fluid dynamics can be deduced with:

$$\nabla \cdot U = 0$$

$$\frac{\partial w}{\partial t} + U \cdot \nabla w = \nu \nabla^2 w + f$$

$$w|_{t=0} = w_0,$$

where $U = (u, v)$ is a velocity vector in 2D field, $w = |\nabla \times U| = \frac{\partial u}{\partial y} - \frac{\partial v}{\partial x}$ is the vorticity, $w_0 \in \mathbb{R}$ is the initial vorticity at $t = 0$. In this dataset, viscosity $\nu$ is set as $10^{-5}$ and the resolution of the 2D field is $64 \times 64$. Each generated sample contains 20 successive time steps, and the task is to predict the future 10-time steps based on the past 10 time steps. The training dataset contains 1000 samples, while the testing dataset contains 100. In the following, we describe two training setups.

**Two training setups**

- **Original training setup:** Samples consist of 20 sequential time steps, with the goal of predicting the subsequent 10 time steps from the preceding 10. Using the roll-out prediction approach as described by Li et al. (2020), the neural operator uses the initial 10-time steps to forecast the immediate next time step. This predicted time step is then merged with the prior 9 time steps to predict the ensuing time step. This iterative process continues to forecast the remaining 10 time steps.

- **New training setup:** Our findings indicate that an amalgamation of deep learning strategies is pivotal for the optimal performance of time-dependent tasks. While Li et al. (2020) employed the previous 10-time steps as inputs for the neural operator, our approach simplifies this. We find that leveraging just the current step's data, similar to traditional numerical solvers, is sufficient. During training, we avoid model unrolling. Instead, from the initial 1000 samples consisting of 20 sequential time steps, we derive 19,000 samples with pairs

of sequential time steps. These are then shuffled and used to train the neural operator to predict the subsequent time step based on the current one. For testing, we revert to the roll-out prediction method as only the initial time step's ground truth is available. These methods align with some techniques presented in Brandstetter et al. (2022). As shown in Table 2, the second approach provides better performance.

For both setups, we use the $L^2$ loss function, the Adam optimizer, and the OneCycleLR scheduler with a cosine annealing strategy. The learning rate starts with $1 \times 10^{-3}$ and decays to $1 \times 10^{-5}$. All models were trained for 500 epochs.

**Pipe**    This dataset in Li et al. (2022) is devoted to simulate incompressible flow through a pipe. The governing equations are:

$$\nabla \cdot U = 0$$
$$\frac{\partial U}{\partial t} + U \cdot \nabla U = f - \frac{1}{\rho}\nabla p + \nu \nabla^2 U.$$

The dataset is generated in the pipe-shaped structured mesh with the resolution of $129 \times 129$. For experiments, we adopt the mesh structure as the input data, and the output is the horizontal fluid velocity within the pipe.

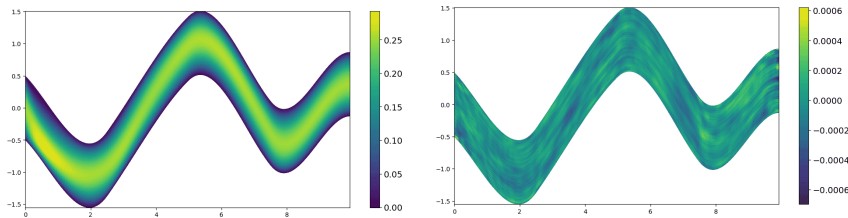

Figure 7: The pipe task: On the left, the ground truth of the pipe flow is displayed, whereas the right illustrates the prediction error of the MgNO.

### D.3    HELMHOLTZ EQUATION

Helmholtz equation in highly heterogeneous media is an example of multiscale wave phenomena, whose solution is considerably expensive for complicated and large geological models. We adopt the setup from (De Hoop et al., 2022), for the Helmholtz equation on the domain $D = [0, 1]^2$. Given frequency $\omega = 10^3$ and wave speed field $c : \Omega \to \mathbb{R}$, the excitation field $u : \Omega \to \mathbb{R}$ solves the equation

$$\begin{cases} \left(-\Delta - \dfrac{\omega^2}{c^2(x)}\right)u = 0 & \text{in } \Omega, \\[2mm] \dfrac{\partial u}{\partial n} = 0 & \text{on } \partial\Omega_1, \partial\Omega_2, \partial\Omega_4, \\[2mm] \dfrac{\partial u}{\partial n} = 1 & \text{on } \partial\Omega_3, \end{cases}$$

where $\partial\Omega_3$ is the top side of the boundary, and $\partial\Omega_{1,2,4}$ are other sides. The wave speed field is $c(x) = 20 + \tanh(\tilde{c}(x))$, where $\tilde{c}$ is sampled from the Gaussian field $\tilde{c} \sim \mathcal{N}(0, \left(-\Delta + \tau^2\right)^{-d})$, where $\tau = 3$ and $d = 2$ are chosen to control the roughness. The Helmholtz equation is solved on a $100 \times 100$ grid by finite element methods. We aim to learn the mapping from $c \in \mathbb{R}^{100 \times 100}$ to $u \in \mathbb{R}^{100 \times 100}$ as shown in Figure 8.

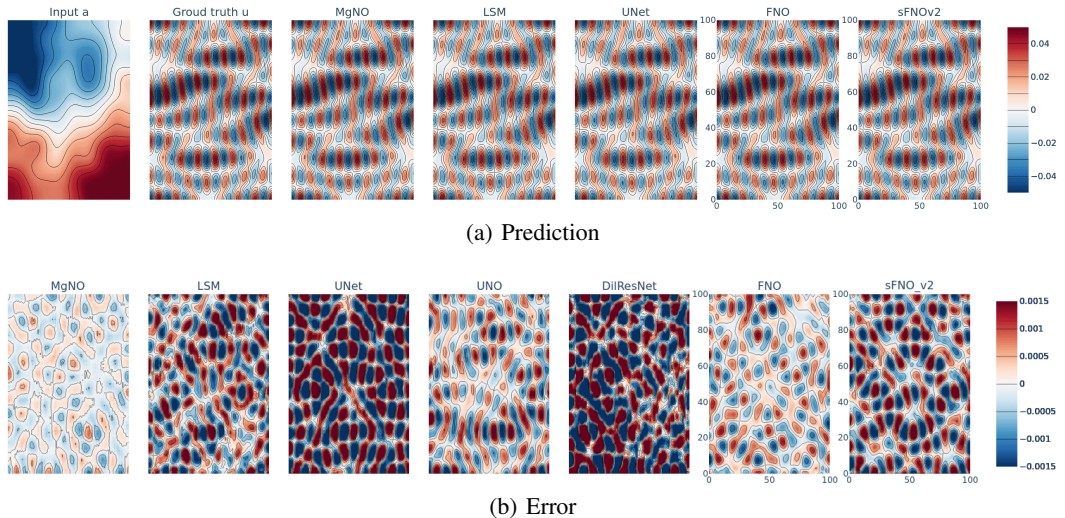

(a) Prediction

(b) Error

Figure 8: The Helmholtz benchmark.

The Helmholtz equation is notoriously difficult to solve numerically. One reason is the so-called resonance phenomenon when the frequency $\omega$ is close to an eigenfrequency of the Helmholtz operator for some particular wave speed $c$. We found that it was necessary to use a large training dataset of size 4000 examples. The test dataset contained 400 examples. All models were trained for 100 epochs.

## E  DISCRETIZATION INVARIANCE

FNO intrinsically leverages the Fourier basis, allowing models trained on lower-resolution datasets to seamlessly process high-resolution inputs. By incorporating appropriate finite element basis functions, MgNO, compatible with the finite element method, demonstrates an equivalent capability. MgNO can train on lower-resolution datasets and evaluate on higher resolutions without the necessity of high-resolution training data, thus accomplishing zero-shot super-resolution.

| Train \ Test | FNO | | | MgNO | | |
|---|---|---|---|---|---|---|
| | 128 | 256 | 512 | 128 | 256 | 512 |
| 64 | 5.2808 | 7.9260 | 9.1054 | 1.032 | 1.178 | 1.205 |

Table 6: Comparison of discretization invariance property for MgNO and FNO for the Darcy multi-scale benchmark. The relative $L^2$ error ($\times 10^{-2}$) with respect to the reference solution on the testing resolution is measured.

## F  MODEL CONFIGURATIONS

We detail the primary configurations of the models in this section. Taking the NUMBER OF ITERATIONS PER LEVEL as an example, the format [[1,1],[1,1],[1,1],[1,1],[1,1],[2]] is adopted. In the context of the first level, the representation [1, 1] implies one a priori iteration followed by one post iteration (as indicated by the two values separated by a comma).

| Modules | | Darcy (smooth/rough/multiscale) | Navier Stokes | Helmholtz |
|---|---|---|---|---|
| | NUMBER OF LEVELS: | 6 | 5 | 6 |
| | CHANNELS PER LEVEL: | [24,24,24,24,24,24] | [32,32,32,32,32] | [32,64,128,256,512] |
| | ITERATIONS PER LEVEL: | [[1,1],[1,1],[1,1],[1,1],[1,1],[2]] | [[1,0],[1,0],[1,0],[2,0],[2]] | [[1,0],[1,0],[1,0],[1,0],[2]] |
| $\mathcal{W}_{Mg}$ : | CONVOLUTION A: | $3 \times 3$, ZEROS PADDING 1 | $3 \times 3$, CIRCULAR PADDING 1 | $3 \times 3$, REFLECT PADDING 1 |
| | CONVOLUTION B: | $3 \times 3$, ZEROS PADDING 1 | $3 \times 3$, CIRCULAR PADDING 1 | $3 \times 3$, REFLECT PADDING 1 |
| | CONVOLUTION R: | $3 \times 3$, ZEROS PADDING 1, STRIDE 2 | $3 \times 3$, CIRCULAR PADDING 1, STRIDE 2 | $3 \times 3$, NO PADDING, STRIDE 2 |
| | CONVTRANSPOSE P: | $4 \times 4$, NO PADDING, STRIDE 2 | $4 \times 4$, NO PADDING, STRIDE 2 | $4 \times 4$, NO PADDING, STRIDE 2 |
| | LAYER NORMALIZATION: | No | YES | YES |
| NUMBER OF LAYERS: | | 4/4/5 | 5 | 3 |
| ACTIVATION FUNCTION: | | GELU | GELU | GELU |

Table 7: Model Configurations

