# OpenReview forum: "MgNO: Efficient Parameterization of Linear Operators via Multigrid"
_ICLR.cc/2024/Conference — ICLR 2024 poster_

### Official Review · Reviewer_4o6c · 2023-10-31

**Soundness:** 3 good
**Presentation:** 3 good
**Contribution:** 3 good
**Rating:** 6
**Confidence:** 3

**Summary:**

In this paper, the authors introduce a novel formulation for neural operators and establish (in Theorem 3.1) the corresponding universal approximation property.

The authors then propose a multi-grid-based parameterization of linear operators, called MgNO, that can be efficiently parameterized and naturally accommodates various boundary conditions.

Numerical results are provided on several popular PDEs, including Darcy, Helmholtz, and Navier-Stokes equations, with different boundary conditions, showing the superiority in prediction accuracy and efficiency of the proposed approach.

**Strengths:**

The paper provides novel insights and novel methodology of neural operators.
The numerical results on different popular PDEs with various boundary conditions look compelling.

**Weaknesses:**

I do not see particular weakness for this paper. See below for some comments.

**Questions:**

I do not have specific questions but the following general comments for the authors:

1. just being curious, can something similar to Theorem 3.1 be said about deep networks as defined in Section 3.2? How the network depth may play a role here?
2. it would be helpful to discuss also the limitation of the proposed approach.

---

> ### Author Response · Authors · 2023-11-15
>
> ## Response to Questions:
>
> **Question 1**:
> Thank you for raising this interesting question. Indeed, Theorem 3.1 pertains solely to shallow neural operators. The theoretical impact of deep neural operator architecture on performance, as well as the role of depth in neural operators, remain open questions.
>
> In fact, the role of depth in traditional deep Neural Networks is not yet fully understood. A vast amount of literature has explored this topic, primarily focusing on the following aspects:
> 1. Expressivity: 2. Approximation power; 3. Implicit bias and regularization effects; 4. The landscape of loss and its impact on training.
> I conjecture that depth in neural operators might play similar roles as in traditional neural networks. However, no rigorous theoretical framework has yet been established to confirm this.
>
> In our experiments, we observed a rapid increase in performance when the depth of MgNO was increased from one hidden layer to 4, 5, or 6 layers. Beyond this point, however, the benefits of increased depth diminish, weighed against the costs of computational complexity and training challenges.
>
> From both theoretical and practical standpoints, whether a significantly deeper neural operator architecture, such as one with 100 layers, can substantially enhance neural operator performance remains an open question.
>
>
> **Question 2**:
> Thank you for drawing attention to this aspect. We acknowledge the need for a discussion on limitations or drawbacks in our concluding remarks section. Accordingly, we incorporate a discussion on the limitations or drawbacks there. One limitation of the current implementation of MgNO is its restriction to inputs and outputs on uniform meshes; otherwise, mesh data has to be integrated into the input such that the input is akin to pixel grids in image data. To address this, we plan to integrate ideas from algebraic multigrid methods to
> enhance our current MgNO framework. This enhancement will enable it to handle inputs and outputs on more
> general meshes, and even extend to arbitrary sampling data points or point clouds.

---

### Official Review · Reviewer_RPM7 · 2023-10-31

**Soundness:** 3 good
**Presentation:** 3 good
**Contribution:** 3 good
**Rating:** 6
**Confidence:** 2

**Summary:**

This paper proposes a new neural operator architecture that does not require lifting and projecting operators. It uses multigrid structures (V-cycle multigrid) to parameterize the linear operators between neurons. Then this paper proves the universal approximation of their proposed parameterization and uses experiments to show the efficiency and accuracy of the method.

**Strengths:**

The neural operator proposed in this paper uses the multi-scale method to get a better parameterization of the weights. It is applied to the spatial domain hence it seems like it won't significantly increase the complexity. From the experimental results, MGNO works quite well compared to other models, which provides convincing evidence that MGNO can be useful.

**Weaknesses:**

1. I am confused about the discretization of the input. It seems like the input still needs to be discretized (Eq.(6)) and the multigrid is applied to the spatial domain. Then I suppose the performance and the efficiency of MGNO are affected by the mesh size $h=1/d$. The performance of MGNO with respect to $d$ is missing in the paper.

2. What about the parameterization with multi-channel inputs? Can this V-cycle be applied to multi-channel inputs? If there is no easy extension, I believe the significance of this work is compromised.

**Questions:**

1. What is the dependence of $n$ with $\epsilon$ in Theorem 3.1?
2. Why are the approximation capabilities so important? It might be a necessary condition for neural operators to learn but is definitely not sufficient. In my understanding, FNO is good at working with different resolutions which probably is not caused by its approximation capability.

---

> ### Author Response · Authors · 2023-11-14
>
> Thank you for your feedback on our paper. We appreciate your recognition of the strengths of our proposed Multigrid Neural Operator (MgNO) architecture and your comments on areas needing clarification. Below, we address each of your concerns and questions.
>
> ### Addressing Weaknesses:
>
> 1. **Discretization of Input**:
>
>  - **Clarification**:
>
> You are correct that the input requires discretization, as shown in Eq.(6). MgNO applies multigrid structures to the spatial domain, which inherently involves a discretization process. For irregular domains, the mesh structure data has to be further integrated into the input data.
>    - **Impact of Mesh Size**:
>
> The performance and efficiency of MgNO are influenced by the mesh size. While discretization invariance, as observed in FNO, suggests that a model's performance is not sensitive to input resolutions for certain tasks, it's important to recognize the limitations of this concept. Discretization invariance can result in uniformly good or uniformly bad performance across different input resolutions. The uniformly good scenario implies that the underlying operator is low-dimensional, where coarse information suffices to capture its behavior. However, this is not typically the case in complex problems, such as fluid flow at high Reynolds numbers or wave propagation at high wave numbers. For instance, in turbulent flows, fine resolution is crucial to accurately represent small-scale phenomena, and failure to capture these details can significantly alter long-range predictions. Thus, expecting uniform performance across various resolutions is unrealistic, particularly for complex real-world problems.
>
> To illustrate this, experiments in Table 6 of Appendix D, compare the performance of MgNO and FNO across different mesh sizes. These experiments reveal that for tasks where fine-scale information is crucial, both models exhibit deteriorated performance compared to the results shown in Table 1. This demonstrates the importance of resolution in such complex problems.
>
> Furthermore, in the study "Representation Equivalent Neural Operators: a Framework for Alias-free Operator Learning," it is proven that discretization invariance is achievable only within a band-limited function space (low dimensional space), which is a rather restrictive condition. Our experiments, therefore, provide valuable insights into the practical implications of mesh size on model performance.
>
> 2. **Parameterization with Multi-Channel Inputs**:
>
> We appreciate your query regarding the handling of multi-channel inputs in our model. This aspect, focusing on both expressivity and complexity, is thoroughly discussed in Section 4.1 of our manuscript.
>
> In our experiments, the model indeed utilizes a multi-channel parametrization, and notably, the input for the Navier-Stokes task is multi-channel, demonstrating the model's capability to handle such data. This approach aligns with classical multigrid methods applied to solve elliptic partial differential equations (PDEs), including linear elasticity problems in 2D or 3D scenarios. In such cases, the V-cycle multigrid methods can be naturally represented as linear MgNO with multi-channel inputs.

---

> > ### Author Response · Authors · 2023-11-14
> >
> > ### Addressing Questions:
> >
> >  **Question 1**:
> >
> > Thanks for this comment. Indeed, achieving qualitative approximation results in relation to the number of neurons for operator approximation would be highly beneficial. In our proof presented in Appendix A, the approximation error is primarily attributed to the first three steps: the piecewise constant (functional) approximation of a continuous operator within a compact set, the parameterization (or approximation) of $\mathcal X$ into finite dimensions for discretizing each functional, and the universal approximation capabilities of classical neural networks with a single hidden layer.
> >
> > For the latter two steps, it is possible to obtain qualitative results under specific assumptions about $\mathcal X$ and the activation functions. However, as of now, there is no qualitative result available for the first step of approximation. Essentially, the approximation power in the first step is obtained by the compactness and the partition of unity over the compact set $\mathcal C \subset \mathcal X$. Some key references include:
> >
> > 1. Kriegl, Andreas, and Peter W. Michor. The Convenient Setting of Global Analysis. Vol. 53. American Mathematical Soc., 1997.
> >
> > 2. Schmeding, Alexander. An Introduction to Infinite-Dimensional Differential Geometry. Vol. 202. Cambridge University Press, 2022.
> >
> > This issue is not unique to our MgNO architecture but is also present in other neural operator frameworks, such as FNO. Within the framework for establishing the approximation results in Appendix A, one potential approach is to impose additional assumptions on the compact set of neural operators or to assume a specific form of the operator to be approximated. This would allow for a more quantitative analysis of the finite approximation by functionals.
> >
> > **Question 2**:
> >
> > You are correct. While the approximation capabilities in operator learning are crucial, they are not the only aspect of concern. We would like to highlight two points in this regard.
> >
> > Firstly, regarding the generalization error, different architectures for neural operators may impart implicit biases or regularization effects, particularly for specific operator learning tasks. In Section 5 of our paper, which discusses Training Dynamics, and in Figures 3 \& 5, we demonstrate that our model exhibits robust generalization capabilities on unseen data. This performance markedly contrasts with models like FNO and UNO. We attribute this to our multigrid-parametrization approach for linear operators. We plan to delve deeper into this phenomenon and provide more comprehensive insights and analysis in our future work.
> >
> > Secondly, our current focus is primarily on the universal approximation properties of continuous operators within a compact set. However, if additional information and characteristics of the operator to be approximated are known, it is certainly possible to propose a more optimized neural operator architecture. For instance, in the context of Darcy's problem, where the operator maps the coefficient function
> > $a(x)$ to the solution function $u(x)$, we know that the operator can be infinitely differentiable or even analytic, given certain assumptions about the domain and the right-hand term. This knowledge could lead to a more suitable architecture for this specific task.
> >
> > Regarding the performance of FNO at different resolutions, I agree with your observation. The key factor is not merely the approximation. In my view, neural operators like the Fourier Neural Operator, Wavelet Neural Operator, and Laplace Neural Operator essentially represent low-rank approximations (truncated to low-frequency modes) of kernel functions in various spectral domains. The so-called ``discretization invariance" emerges naturally due to the transformation, which is well-established and does not need to be trained, from spatial to spectral domains. However, this comes with a trade-off: truncating to low-frequency modes in spectral domains can lead to instability in the architectures when dealing with inputs/outputs that contain (or are dominated by) high-frequency modes. The numerical results in Table 6 in Appendix E in our paper and in the following reference may offer some insight into this:
> >
> > Lu, Lu, et al. "A comprehensive and fair comparison of two neural operators (with practical extensions) based on fair data." Computer Methods in Applied Mechanics and Engineering 393 (2022): 114778.

---

> > > ### Comment · Reviewer_RPM7 · 2023-11-20
> > >
> > > Thank the authors for addressing my concerns. I have increased the score.

---

### Official Review · Reviewer_H6UK · 2023-11-01

**Soundness:** 3 good
**Presentation:** 3 good
**Contribution:** 3 good
**Rating:** 6
**Confidence:** 2

**Summary:**

The authors present a concise neural operator architecture designed for operator learning. This architecture draws inspiration from a conventional fully connected neural network, with each neuron in a nonlinear operator layer being determined by a bounded linear operator connecting input and output neurons, in addition to a function-based bias term. To address the efficient parameterization of these bounded linear operators within the architecture, the authors introduce MgNO (Multigrid Neural Operator). MgNO utilizes multigrid structures to effectively model and approximate these linear operators, eliminating the need for conventional lifting and projecting operators while accommodating diverse boundary conditions. Numerical experiments highlight the advantages of MgNO over CNN-based models and spectral-type neural operators, demonstrating improved ease of training and reduced susceptibility to overfitting.

**Strengths:**

1. The authors introduced a novel formulation for neural operators that characterizes neuron connections as bounded linear operators within function spaces. This eliminates the need for traditional lifting and projecting operators, simplifying the architecture.

2. The MgNO architecture, which leverages multigrid structure and multi-channel convolutional form, efficiently parameterizes linear operators while accommodating various boundary conditions. This approach enhances both accuracy and efficiency.

3. Empirical evaluations demonstrate superior performance of MgNO across multiple partial differential equations (PDEs), including Darcy, Helmholtz, and Navier-Stokes equations. This indicates the effectiveness and versatility of the proposed approach.

**Weaknesses:**

The limitations and drawbacks of the proposed methods are not explicitly mentioned

**Questions:**

1. The proposed method may not scale well to larger datasets or more complex problems. Any remark for this?

2. Is it possible to extend the universal approximation theorem to encompass general Banach spaces for X and Y?

---

> ### Author Response · Authors · 2023-11-15
>
> ## Addressing Weakness:
>
> Thank you for highlighting this aspect. We acknowledge the need for a discussion on limitations or drawbacks in our concluding remarks section. Accordingly,
> we incorporate a discussion on the limitations or drawbacks there. One limitation of the current implementation of MgNO is its restriction to inputs
> and outputs on uniform meshes; otherwise, mesh data has to be integrated into the input such that the input is akin to pixel grids in image data. To address this, we plan to integrate ideas from algebraic multigrid methods to
> enhance our current MgNO framework. This enhancement will enable it to handle inputs and outputs on more
> general meshes, and even extend to arbitrary sampling data points or point clouds.
>
>
> ## Response to questions
> **Question 1**:
> We thank the reviewer for raising concerns regarding the scalability of our proposed method. The MgNO architecture, central to our approach, is specifically tailored for efficient parameterization of linear operators within function spaces. This is achieved through a multigrid structure and multi-channel convolutional form, enabling scalability to high-resolution inputs typically encountered in complex problems.
>
> Importantly, the multigrid structure of MgNO is recognized for its effectiveness in handling large-scale problems, particularly in numerical solutions to PDEs. This aspect of our architecture underpins our confidence in its scalability.
>
> Empirically, MgNO has demonstrated superior performance in both accuracy and efficiency, as evidenced in our results. Notably, it features the smallest number of parameters as shown in Table 1 of our paper. To directly address the scalability concern, we conducted additional experiments to assess MgNO's performance at various input resolutions -- $64 \times 64$, $128 \times 128$, $256 \times 256$, and $512 \times 512$ -- while monitoring memory usage during both forward and backward passes. For MgNO, the recorded memory usages were 145.04 MB, 574.16 MB, 2290.66 MB, and 9156.64 MB, respectively. In comparison, for the Fourier Neural Operator (FNO), the memory usage were 184.25 MB, 630.16 MB, 2342.80 MB, and 9051.43 MB at the same resolutions. These results indicate a near-linear relationship between memory usage and the square of the resolution $(n^2)$, where $ n$ represents the resolution. This linear trend in resource usage, coupled with the comparative efficiency over FNO, further substantiates our method's scalability.
>
>
> In conclusion, both the architectural design of MgNO and the empirical evidence from our experiments reinforce our confidence in its scalability to more demanding scenarios, thereby addressing the concerns raised.
>
>
>
> **Question 2**:
> Thank you for the insightful point. Indeed, we can extend the universal approximation theorem to encompass more general Banach or Hilbert spaces, which may not necessarily be spaces of functions. As evident from the neural operator architecture in Equation (1), the definitions of $\mathcal W_i$, $\mathcal B_i$, and $\mathcal A_i$ are independent of the structures of $\mathcal{X}$ and $\mathcal{Y}$. This allows for a direct extension to abstract Banach spaces.
>
> However, there are two critical aspects to consider. Firstly, we must assume the existence of at least one Schauder basis, enabling the uniform approximation of $\mathcal{X}$ by finite-dimensional spaces. This assumption is crucial for achieving the approximation rate as outlined in the second step of the original proof in Appendix A.
>
> Secondly, the definition of the activation function needs attention. In the initial version of our manuscript, we proposed a property for the nonlinear activation operator ($\widetilde{\sigma}: \mathcal{Y} \rightarrow \mathcal{Y}$): there exist two elements $\xi, \zeta \in \mathcal{Y}$ such that $\widetilde{\sigma}(a \xi) = \sigma(a) \zeta$, where $\sigma: \mathbb{R} \rightarrow \mathbb{R}$ is a real, non-polynomial function. This approach facilitates a proof analogous to the third step in Appendix A.
>
> With these assumptions for $\mathcal{X}$, $\mathcal{Y}$, and the activation operator $\widetilde{\sigma}$, we can replicate the proof in Appendix A to achieve similar universal approximation results for highly general or abstract Banach spaces $\mathcal{X}$ and $\mathcal{Y}$. In the initial version of our manuscript, we included this result. However, we later decided to omit it, concerned that its generality might lead to misunderstandings among readers.

---

### Official Review · Reviewer_5jaY · 2023-11-06

**Soundness:** 4 excellent
**Presentation:** 3 good
**Contribution:** 3 good
**Rating:** 8
**Confidence:** 4

**Summary:**

This paper proposes a new neural network architecture based on analogies between convnets and the multigrid method (these analogies have been previously identified with an architecture called MgNet). The authors prove a universality theorem for a network whose neurons are operators and use this theorem to motivate a neural operator which is a deep network with neurons being (elliptic) operators implemented via multigrid. Numerical experiments show competitive performance against existing neural operator architectures.

**Strengths:**

- The paper is very clearly written, the ideas are solid and the numerics are strong. I like the spirit of an architecture where neurons become operators: this appears very natural and the right thing to do.

- Using a multigrid (or in general multiscale) representation of operators is also the right thing to do. Similar ideas go back to efficient wavelet approximations of operators of Beylkin, Coifman, and Rokhlin. It is nice to see this done explicitly and clearly it yields strong performance.

- The experiments are very well executed.

**Weaknesses:**

- Many of the ideas might have already been present in MgNet; it would be nice to see a comment.

- It is not clear to me what (9) is stating since you don't state the typical values of c. How should we use / interpret (9)?

- The motivation via exact encoding of boundary conditions is sound but I would also say quite easy to implement with other neural operators. For example the FNO by default works on the torus but via zero padding could be easily adapted to Neumann or Dirichlet boundaries (though not as easily as MgNO).

**Questions:**

- Why is there no comparison with FNO in Figure 7 (Helmholtz)? It might be nice to compare / combine with https://arxiv.org/pdf/2301.11509
- The sentence "Regarding our training setup, we adopted a meticulous approach." is strange.

---

> ### Author Response · Authors · 2023-11-16
>
> ## Addressing Weakness
> **Weakness 1**:
> We appreciate your valuable feedback. For initial remarks on MgNet, please refer to the 'Background and Related Work' section, particularly the subsection titled 'Multigrid'. We include a more comprehensive discussion here and also in the revised manuscript.  MgNet is the pioneering work to introduce multigrid to neural network architecture. The authors rigorously demonstrated that the linear V-cycle multigrid structure for the Poisson equation,  can be represented as a convolutional neural network, despite their experimental focus on nonlinear multigrid structures for vision-related tasks. Our main idea is that the key of operator learning is how to efficiently parametrize a rich family of linear operators integrated with specific boundary conditions. The impressive performance outcomes we've recorded offer robust evidence in favor of this approach.  The insight of MgNet forms a cornerstone in the implementation level of our approach in utilizing multigrid to parameterize such operators.  However, as we have discussed in the "Background and Related Work" section under "Multigrid," the original MgNet and its subsequent adaptations in the context of numerical PDEs and forecasting scenarios, have not provided a simple but effective solution in the realm of operator learning.
>
>
> **Weakness 2**:
> Thank you for highlighting this aspect. We should have added a few more words for this in our paper. The constant $c$ in Equation (9) is determined by the dimensions and properties of $a(x)$ but remains independent of the mesh size $h$. Typically, a classical iterative method (denoted as $ \mathcal{W} $) for solving discrete elliptic PDEs (such as $ A_h u_h = f_h $) will exhibit a convergence rate of $ 1-\frac{1}{c_h} $, where $ c_h = \mathcal{O}(h^{-2}) $ with mesh size (resolution) $ h $. This indicates that the approximation rate of $ \mathcal{W} $ for $ A_h^{-1} $ (i.e., the inverse of discrete elliptic operators, which is the discrete Green's function) is not uniform and deteriorates when $ h $ is small. However, Equation (9) reveals that if we employ the operator implemented by a multigrid iterative method (denoted as $ W_{Mg} $), the convergence rate becomes $ 1-\frac{1}{c} $. This implies that the approximation rate of $ W_{Mg} $ to $ A_h^{-1} $ (Green's function) is uniformly bounded by $ 1-\frac{1}{c} $, independent of the mesh size $ h $. Therefore, Equation (9) demonstrates that $ W_{Mg} $ has significantly enhanced expressivity for general linear operators between two finite element spaces. Specifically, it achieves a uniform approximation rate for a special type of kernel functions, namely, Green's functions, which are the inverses of elliptic operators. This insight is crucial in motivating the use of multigrid methods to parameterize the linear operator $ W_i $.
>
> In Section C of the Appendix, we provide a detailed numerical example to quantify the constant $c$ and approximation rate.
>
> **Weakness 3**:
> We agree that other neural operators can always zeros pad the input to a larger enough (for example padding size 7 for the baseline FNO) domain such that the output functions space is redefined on larger domain. Without this strategy, FNO would be constrained to learning mappings to periodic function space on the target domain. However, it's important to note that padding (always zero-padding for any boundary conditions), is not reflective of an operator's intrinsic ability to learn specific boundary conditions and should be considered more as a workaround than a solution.
>
> In contrast, our distinct parametrization method enables the explicit expression of the multigrid solver.  Consequently, the inverse of  elliptic operators $\mathcal L$ under a linear FEM framework can be approximated by  MgNO operators rigorously. This includes, but is not limited to, the inverse Laplacian $ \Delta^{-1}$ with any boundary conditions.  Furthermore, the constructive approximation of the inverse Laplacian $ \Delta^{-1}$ with any boundary conditions using MgNO explicitly requires the padding mode to be zero, reflect, or periodic (with a padding size of 1), corresponding to Dirichlet, Neumann, and periodic boundary conditions, respectively. This is achieved without the requirement to expand the domain via padding.  In Section C of the Appendix, we have further provided a detailed numerical example illustrating the construction of a linear MgNO operator $\mathcal{W}_{Mg}$, showcasing the integration of boundary conditions into MgNO and its approximation capabilities.  In summary, we assert our approach with MgNO inherently integrates boundary conditions into its structure without resorting to padding to larger domain, offering a more natural and mathematically sound method for handling various boundary conditions.

---

> ### Author Response · Authors · 2023-11-16
>
> ## Response to Questions
> **Question 1**:
> Thank you for your valuable suggestion. In the revised version of our manuscript, we have now included a comparison with FNO in Figure 7 (Helmholtz). Additionally, we have incorporated a comparison with sFNO-v2 as referenced in the paper you mentioned [https://arxiv.org/pdf/2301.11509]. For this comparison, we employed the default hyperparameter settings as provided in the paper, using their official implementation.
>
> However, it's important to note that the results we obtained did not precisely mirror the performance reported in the paper. This discrepancy could potentially be attributed to the hyperparameters not being fine-tuned for the specific task in our study.
>
> | **Model**       | Time | Params(m) | $L^2 (\times 10^{-2})$ | $H^1 (\times 10^{-2})$ |
> |-----------------|------|-----------|------------------------|------------------------|
> | **FNO2D**       | 5.1  | 1.33      | 1.69                   | 9.92                   |
> | **UNet**        | 7.1  | 17.26     | 3.81                   | 23.31                  |
> | **DilResNet**   | 10.8 | 1.03      | 4.34                   | 34.21                  |
> | **U-NO**        | 21.5 | 16.39     | 1.26                   | 8.03                   |
> | **sFNO-v2**     | 30.1 | 12.0      | 1.72                   | 10.40                  |
> | **LSM**         | 28.2 | 4.81      | 2.55                   | 10.61                  |
> | **MgNO**        | 15.1 | 7.58      | **0.71**               | **4.02**               |
>
> _Performance on Helmholtz._
>
> **Question 2**:
> Thank you for your careful review. We have revised the sentence accordingly to "In our training setup, we implemented a comprehensive approach."

---

### Author Response · Authors · 2023-11-20
**Revisions to the manuscript**

We thank all the reviewers' valuable comments to improve the quality and clarity of our paper. In response, we have made the following major revisions to the manuscript, with all changes highlighted in blue:

- A new Section C has been added to the Appendix, presenting a concrete numerical example. This section illustrates the integration of boundary conditions into MgNO and quantifies the constant $c$ in Equation (9) along with the approximation rate numerically.
- An expanded discussion on the pioneering work MgNet has been included.
- The Helmholtz task now features a new baseline, sFNO-v2, and a comparison with FNO is provided in Figure 7 (Helmholtz).
- The discussion on the multi-channel parametrization of MgNO layers has been relocated to Section 4.2, with sentences rephrased for improved clarity and coherence. Details on tasks involving multi-channel input have been added to Section D in the Appendix.
- An expanded discussion on the limitations of our approach has been incorporated.

---

### Meta-Review · Area_Chair_uqSH · 2023-12-13

**Metareview:**

The paper introduces MgNO, a new architecture for parametrizing neural operators. Neural operators parametrize a solution operator between a space of PDEs, and their solutions --- so they need to map a function to a function. To make the problem "finite dimensional" and parametrizable by a neural network, typical architectures "project down" to a final dimensional space (with a learned encoder), and "project up" (with a learned decoder). This architecture instead considers a neural network parametrizing the operator, such that each neuron parametrizes an operator. As in a usual neural net, a neuron operator is a linear combination of neighbor linear operators, followed by a nonlinearity.

In some sense, the approach is a merge of pre-existing ideas, so there's nothing substantively new. The theoretical results on universal approximation also follow from fairly standard results on universal approximation in classical settings about approximating neural networks. Finally, even though there is no encoder/decoder, the method is still sensitive to the *grid* discretization. On the other hand, the method has clean benefits compared to competitor methods (e.g. versions of FNO) both in terms of parameter count / runtime, and L2/H1 performance on some simple PDEs like (rough) Darcy.

**Justification For Why Not Higher Score:**

The paper's major ideas are relatively straightforward and combine pre-existing ideas, as are the theoretical results. The empirical benchmarking is done on a relatively small number of PDEs, and with very regular grids for the inputs.

**Justification For Why Not Lower Score:**

The paper presents a clean and natural idea, and demonstrates superior performance on some commonly tested simple PDEs like (rough) Darcy.

---

### Decision · Program_Chairs · 2024-01-16

Accept (poster)